# Modern Development and Production of a New Live Attenuated Bacterial Vaccine, SCHU S4 *ΔclpB*, to Prevent Tularemia

**DOI:** 10.3390/pathogens10070795

**Published:** 2021-06-23

**Authors:** J. Wayne Conlan, Anders Sjöstedt, H. Carl Gelhaus, Perry Fleming, Kevan McRae, Ronald R. Cobb, Roberto De Pascalis, Karen L. Elkins

**Affiliations:** 1Department of Human Health and Therapeutics, National Research Council-Canada, Ottawa, ON K1A 0R6, Canada; perry.fleming@nrc-cnrc.gc.ca (P.F.); kevan.mcrae@nrc-cnrc.gc.ca (K.M.); 2Laboratory for Molecular Infection Medicine Sweden (MIMS), Department of Clinical Microbiology, Umeå University, SE-901 85 Umeå, Sweden; anders.sjostedt@climi.umu.se; 3MRIGlobal, Kansas City, MO 64110, USA; cgelhaus@mriglobal.org; 4Ology Bioservices Inc., Alachua, FL 32615, USA; ron.cobb@resilience.com; 5US Food and Drug Administration, Center for Biologics Evaluation and Research, Silver Spring, MD 20993, USA; roberto.depascalis@fda.hhs.gov (R.D.P.); karen.elkins@fda.hhs.gov (K.L.E.)

**Keywords:** tularemia, *Francisella tularensis*, live attenuated vaccine, product development

## Abstract

Inhalation of small numbers of *Francisella tularensis* subspecies *tularensis* (*Ftt*) in the form of small particle aerosols causes severe morbidity and mortality in people and many animal species. For this reason, *Ftt* was developed into a *bona fide* biological weapon by the USA, by the former USSR, and their respective allies during the previous century. Although such weapons were never deployed, the 9/11 attack quickly followed by the Amerithrax attack led the U.S. government to seek novel countermeasures against a select group of pathogens, including *Ftt*. Between 2005–2009, we pursued a novel live vaccine against *Ftt* by deleting putative virulence genes from a fully virulent strain of the pathogen, SCHU S4. These mutants were screened in a mouse model, in which the vaccine candidates were first administered intradermally (ID) to determine their degree of attenuation. Subsequently, mice that survived a high dose ID inoculation were challenged by aerosol or intranasally (IN) with virulent strains of *Ftt*. We used the current unlicensed live vaccine strain (LVS), first discovered over 70 years ago, as a comparator in the same model. After screening 60 mutants, we found only one, SCHU S4 *ΔclpB*, that outperformed LVS in the mouse ID vaccination-respiratory-challenge model. Currently, SCHU S4 *ΔclpB* has been manufactured under current good manufacturing practice conditions, and tested for safety and efficacy in mice, rats, and macaques. The steps necessary for advancing SCHU S4 *ΔclpB* to this late stage of development are detailed herein. These include developing a body of data supporting the attenuation of SCHU S4 *ΔclpB* to a degree sufficient for removal from the U.S. Select Agent list and for human use; optimizing SCHU S4 *ΔclpB* vaccine production, scale up, and long-term storage; and developing appropriate quality control testing approaches.

## 1. Introduction

Tularemia is the generic term for a spectrum of infectious diseases caused by the facultative intracellular bacterium, *Francisella tularensis* subspecies tularensis (*Ftt*), or *F. tularensis* subspecies holarctica (*Fth*) [1,2]. *Ftt* is only found in North America, while *Fth* is found throughout the Northern hemisphere. Tularemia is uncommon in the USA, but certain European countries and especially Sweden experience hundreds to thousands of cases of *Fth* infection annually [3,4]. Disease symptoms range from local eschar formation and lymphadenopathy, to high mortality following fulminant systemic infection that disseminates to lungs, liver, spleen, and blood, particularly after inhalation of *Ftt* small particle aerosols [5]. 

Although *Francisella* infects a wide range of animals as well as humans [1], no cases of human-to-human transmission have ever been convincingly documented. Nonetheless, despite the lack of contagiousness, *Ftt*’s virulence made it a primary target for bioweapon development between the 1940s and 1980s [2,6,7]. To counter the threat of biowarfare using *Ftt*, beginning in the 1940s, several vaccines were developed, ranging from so-called “Foshay” killed bacterial vaccines to the live vaccine strain (LVS) developed in the USA from live attenuated strains used in the USSR [8,9,10]. Non-human primate (NHP) experiments, field observations, and human vaccination and challenge studies have suggested that both killed and live vaccines protected against dermal exposure to virulent *Ftt* [11,12,13,14,15,16]. However, only LVS, administered by skin scarification, alleviated infections due to *Ftt* inhalation; but protection was suboptimal and could be overwhelmed by exposure to larger challenge doses [16]. Efficacy improved if LVS was administered by aerosol, but aerosol vaccination was accompanied by substantial reactogenicity [17]. Despite the drawbacks, large amounts of LVS have been produced under contract for the U.S. Department of Defense and studied as an investigational vaccine [18], but LVS remains unlicensed.

Interest in tularemia vaccines declined as the Cold War waned and bioweapon treaties were enacted. However, the 9/11 terrorist attack, followed quickly by the Amerithrax attack via the U.S. mail, rapidly rekindled interest in the development of effective countermeasures against potential biothreat agents, including *Ftt* [19,20]. From 2003 to 2010, U.S. spending on biodefense spurred significant activity to develop a new tularemia vaccine, and a variety of approaches have been explored as reviewed previously [21,22,23]. We hypothesized that a rationally attenuated vaccine based on a *Ftt* strain, such as SCHU S4, might outperform *Fth*-based LVS in terms of both safety and efficacy against *Ftt* threats [24,25]. We previously reported developing new methods to generate unmarked deletion mutants of *Ftt* SCHU S4 [26], first, resulting in defined mutants that lacked either *FTT0918* (now called *fupA*) or *FTT0919* (*fupB*) genes [24]. These were targeted because LVS, as well as another attenuated strain, *Ftt* FSC043, previously designated as SCHU AV [24], contained a genetic lesion spanning these two genes; this mutation was subsequently shown to be responsible for most of the attenuation of LVS [27]. However, *ΔFTT0919* was not overtly attenuated for BALB/c mice, and *ΔFTT0918* was only moderately attenuated as compared with LVS and FSC043.

Therefore, we elected to generate and screen many mutant strains, with the goal of identifying another single gene deletion mutant of *Ftt* SCHU S4 that was at least as attenuated as LVS in mice but more effective against respiratory challenge with virulent bacteria. The screening approach is outlined in Figure 1. Briefly, each mutant’s virulence was evaluated by administering 10^3^ colony forming units (CFU) ID, a route chosen to mimic scarification used to vaccinate humans with LVS. If mice displayed no overt disease at 10^3^ CFU ID, additional mice were immunized ID with 10^5^ or 10^7^ CFU. All mice surviving vaccination were subsequently challenged ID or by aerosol with virulent *Ftt* FSC033 or SCHU S4.

In all, 60 gene deletion mutants, including 57 individual gene deletion mutants and three mutants spanning several genes, were generated and screened (Table 1). The virulence properties of most of these mutants have been previously reported by us [24,25,28,29,30,31,32,33,34,35,36]. Intradermal administration of several of the most highly attenuated SCHU S4 mutants, e.g., *Δwbtc*, *ΔgplX*, and *ΔclpB*, as well as FSC043 and LVS, protected BALB/c mice from a subsequent 1000 CFU ID challenge with *Ftt* SCHU S4 [29]. However, only *Ftt* SCHU S4 *ΔclpB* (hereafter, *ΔclpB*), given ID, protected BALB/c mice against aerosol or IN challenge with up to 100 CFU of SCHU S4 [37,38]. Of note, *ΔclpB* was generated by deletion of 2463 out of 2580 bp of the *ΔclpB* gene, and complementation restored a wild-type level of virulence. Furthermore, deleting other genes from Columns 1–3 in Table 1 from *ΔclpB* to yield double mutants always led to a significant decrease in protection against IN *Ftt* challenge of BALB/c mice [25]. Therefore, *ΔclpB* was chosen for additional development.

Because tularemia is not only rare and sporadic in nature, but also can cause high morbidity and mortality, we expect clinical development and licensure in the USA to be pursued via the FDA “Animal Rule.” This regulatory pathway allows for evaluation of efficacy using two animal models, and the approach depends on developing a rational means to bridge the outcomes from animal data to humans [39,40]. Mice are useful for virulence screening and immunological studies, while Fischer 344 rats have advantages for vaccination studies. In particular, rat infections mimic the different infection outcomes seen in humans following *F. novicida* or *Fth* infections (survival of moderate doses) as compared with *Ftt* infections (increased virulence) [41,42,43]. Product characterization studies of *ΔclpB* described here, therefore, have taken advantage of both animal models. Efficacy and immunological correlate studies using Fischer 344 rats and cynomolgus macaques are subjects of manuscripts in preparation, while production and manufacturing-related studies are the subject of the present report.

## 2. Results

### 2.1. Studies Supporting Attenuation and Removal of SCHU S4 ΔclpB from the U.S. Select Agent List

Mutants of fully virulent *F. tularensis*, classified as a Select Agent in the USA, continue to be considered Select Agents themselves (https://www.selectagents.gov/, accessed on 25 May 2021) until evidence to the contrary is considered sufficient for any such mutants to be formally removed from the Select Agent list. In order to develop *ΔclpB* as a vaccine, it is crucial to understand the extent of its attenuation and also to explore characteristics specifically supporting an application to remove it from Select Agent classification. Since LVS has been studied in humans and was excluded from the Select Agent list by describing evidence for its attenuation, we performed a series of experiments comparing the virulence properties of *ΔclpB* to LVS. These experiments are summarized in Table 2. As compared to LVS, *ΔclpB* infection was notably less lethal for BALB/c mice when administered intranasally (IN), produced similar or less skin necrosis and clinical signs in mice, and grew similarly in human and mouse macrophages (not shown). 

An infection of highly immunodeficient SCID mice is often used to evaluate the extent of microbial attenuation, even when the ultimate outcome of infection is death. Here, an infection of SCID mice with *ΔclpB* resulted in longer times to death than those observed after LVS infection [25]. Moreover, only an infection with very large doses of *ΔclpB* led to death of guinea pigs, which, like mice, are highly susceptible to virulent *F. tularensis* [9] (Figure 2).

Then, we evaluated the potential for *ΔclpB* to revert to a virulent phenotype. For these experiments, death of BALB/c mice when given an IN dose of 10^4^ CFU (the lowest LD_50_ reported for *ΔclpB*, Table 2) was considered to be to reflect gain of virulence. *ΔclpB* was passaged in vivo through mice five times, and in vitro ten times on cysteine heart agar supplemented with hemoglobin (CHAH) [24], and stocks prepared for infection after the last passage of each. For *ΔclpB* bacteria passaged either in vivo or in vitro, all BALB/c mice survived IN inoculation at a dose of 10^4^ CFU (Table 3, top half).

To evaluate the potential for recombination, we further evaluated stocks of bacteria in which *ΔclpB* was co-cultured on CHAH agar with a distinct SCHU S4 *ΔiglC* deletion mutant through 5 passages in vitro. All of the mice survived infection with 10^8^ co-cultured bacteria (Figure 3); furthermore, these mice were effectively vaccinated, since all survived when subsequently challenged with fully virulent SCHU S4. Other mice received the same amount of co-cultured bacteria spiked with 50 CFU SCHU S4; about 35% of these mice survived. Normally, ID inoculation of 50 CFU of SCHU S4 alone would result in 100% deaths (mean time to death of 7 days). However, mixing with co-cultured bacteria seems to have somewhat blunted the virulence of the fully virulent bacteria, possibly because the massive excess of the co-cultured bacteria induced concomitant inflammatory immune responses. The combined data above led to the removal of *ΔclpB* from the Select Agent list in 2014 (https://www.selectagents.gov/sat/exclusions/hhs.htm, accessed on 27 May 2021). Subsequently, exclusion was applied to *ΔclpB* carrying any additional mutations (T. Wu, personal communication). 

### 2.2. Clearance of ΔclpB from Fischer Rat Tissues 

As noted, Fischer 344 rats provide a valuable model for testing tularemia vaccines [42,43,45,46,47]. Therefore, we expect rats to be used for testing tularemia vaccines under the “Animal Rule”. Thus, we evaluated distribution and clearance of *ΔclpB* from the spleens of infected male and female rats given ID doses of *ΔclpB* ranging from 10^3^–10^9^ CFU. All rats survived infection, and no bacteria were detected in spleens by 40 days after infection (Figure 4). No significant differences in bacterial burdens were found between male and female rats at any dose on any day. While rats survived ID infection with 10^9^ CFU, 10^10^ CFU ID was uniformly lethal for both sexes (data not shown). 

In a separate experiment, male and female rats were inoculated ID with 30, 300, or 3000 CFU of *ΔclpB*, and splenic burdens were determined on Day 14. One spleen from a female rat inoculated with 30 CFU was sterile. All other rats had a splenic burden of ~4.5 log_10_, with no significant differences among groups (data not shown). Taken together, these data support performing challenge studies of vaccinated rats after Day 30, when vaccinating bacteria are no longer present in tissues.

### 2.3. ΔclpB Vaccine Production Optimization, Scale-Up, and Quality Control Testing

The original clinical lots of LVS, made in the 1960s, and the LVS lots made in the 2000s by the DynPort Vaccine Company (DVC), were both produced by growing LVS in modified casein partial hydrolysate (MCPH) broth [9,18]. At NRC-C, an initial stock of *ΔclpB* was made from a slant growth of the mutant obtained from Umea University. This was expanded by confluent growth on CHAH, and then subsequently harvested and resuspended in freezing medium [24], dispensed into 1.0 ml aliquots, and stored at −80 °C. This is referred to as “original stock” throughout the current manuscript. Then, a working research cell bank was prepared at NRC-C by passaging original *ΔclpB* five times in MCPH broth, and the final passage was used to prepare 1000 × 1.0 mL vials that were stored at −80 °C. This is referred to as “new stock” throughout the current manuscript and served as a research cell bank. Vials of new stock were shared among the vaccine development teams (see authorship for affiliations) and were used as the starting material for all subsequent vaccine development studies reported herein and elsewhere. The characteristics of new stock *ΔclpB* are listed in Table 4 as compared with “original stock” (i.e., before passage through MCPH broth); all were considered to be representative of results to date and acceptable as bacterial source stocks for future work. Subsequently, a cGMP master cell bank was prepared at a commercial manufacturing organization, Ology Bioservices, from a single isolated colony, frozen in freezing medium in 1 mL vials, and characterized, as described in Appendix A. 

A major hurdle in vaccine development is the ability to scale-up production to levels suitable for the target population. Previously, LVS growth for extraction of *F. tularensis* LPS in a 30-liter BSL3 fermenter (working volume 22 L) at NRC-C [48] resulted in yields of 10^14^ CFU, an amount sufficient for ~10 million ID human doses or 1 million scarification doses, assuming a dose similar to LVS [49], the difference reflecting the wastage inherent to the latter administration route. Therefore, we evaluated growth approaches amenable to a ~22 L scale and which were appropriate for GMP production. First, we explored growth in Chamberlain’s defined broth (CDB), since all ingredients are available in a highly purified form and not sourced from cows (minimizing the risk of contamination by bovine spongiform encephalopathy prions). Because previous reports [50] indicated that LVS grown in CDB increased in virulence for mice, we tested the virulence of *ΔclpB* grown to 22 L scale in CDB. The results showed that fewer mice infected IN with *ΔclpB* either passaged in CDB or grown at scale in CBD survived infection with 10^4^ CFU as compared with the starting stocks grown at lab scale (Table 3, bottom half). Although the significance of the subtle change in *ΔclpB* when grown in CDB is not clear, we elected to evaluate growth of *ΔclpB* in MCPH, the medium used to develop the original human lots of LVS and later used by DVC. We produced three 22 L fermenter batches at NRC-C at 1–2-year intervals, the latter two using *ΔclpB* research cell bank stocks. The outcomes of fermentation at scale appeared to be highly reproducible in terms of yield from the starter culture flasks, yields from fermenter growth, bacterial doubling time, and biomass (Table 5). The *ΔclpB* growth curve from the second NRC-C fermenter run (Table 5) as monitored by OD_600_, illustrates growth in MCPH. During fermenter runs two (Figure 5A) and three (not shown), *ΔclpB* cultures were sampled at various intervals which represented late log phase, early stationary phase, and mid-stationary phase. On the basis of the results from testing these samples in putative lot release and biological characterization assays (see below), we considered the early mid-stationary phase to be the optimal time period for harvest. Approximately 200 mL/time point of fermenter growth was sufficient for R&D purposes at NRC-C. Fermenter contents collected at the end of NRC-C fermenter Run 1 (Table 5) were used for long-term preservation studies. To mimic large-scale diafiltration prior to lyophilization, ~100 mL was centrifuged at 6700× *g* for 20 min at +4 °C. The pellet was washed once in lyophilization medium and resuspended in 50 mL of the same, and then aliquoted into 5 mL lyovials in 2 mL volumes (concentrated stock). Additionally, a 1/1000 dilution of the concentrated stock was made in lyophilization medium and dispensed as above for freeze-drying. The LVS vaccine produced in the 1960s contained 50 doses/vial, whilst that produced by DVC contained a single (2 × 10^7^ CFU) dose. Therefore, we attempted to mimic both of these situations with *ΔclpB.* For all other purposes, sucrose was added to 10% *w/v* to the MCPH growth, aliquoted in 1.0 mL volumes, and frozen at −80 °C until required. The bulk of the fermenter growth was harvested by centrifugation, and the cell paste stored in 50–200 g amounts and frozen at −80 °C.

SOPs developed during fermenter Run 2 were transferred to Ology Biosciences. Several process development runs were performed to demonstrate the reproducibility of the procedure developed at NRC-C. As shown in Figure 5B,C, the growth patterns of *ΔclpB* in two difference process runs were nearly identical, and very similar to Run 2 at NRC-C (Figure 5A). Then, a 25 L scale manufacturing run was performed. The product from the initial 25 L run was characterized, and the bulk product conformed to expected characteristics (Appendix A). The products generated from these runs were then diluted in lyophilization buffer to a target concentration of 5 × 10^8^ CFU/mL and subsequently lyophilized (see below). The final drug product was then characterized using a panel of working lot release assays, as described in Table 6.

### 2.4. Long-Term Storage of ΔclpB 

The LVS produced in the 1960s was lyophilized and frozen at −80 °C at a concentration of 5 × 10^10^ CFU; the original lyophilized vials have been used for in vitro, preclinical, and clinical studies up to the present [9,52,53,54]. This experience with successful long-term storage of LVS was the basis for testing similar storage conditions for *ΔclpB*. The original vaccine lots and the newer lots produced by DVC both used 10 mM phosphate buffer containing 10% *w/v* sucrose and 1.3% gelatin as the lyophilization medium [9,18]. In the former case, a 90% loss in viability, measured as a decline in CFU, was associated with lyophilization and reconstitution in water for injection, whereas DVC did not report similar data in published work. Others empirically tested multiple other matrices for foam-drying of LVS [55], the best of which showed no loss of viability after twelve weeks of storage at 25 °C. The lyophilization process described by Eigelsbach and Downs [9] and Pasetti et al. [18], when applied to *ΔclpB*, resulted a in viability loss of >99% (not shown). Furthermore, we did not have access to the foam drying apparatus used by Ohtake et al. [55]. Therefore, we empirically tested lyophilization formulations comprised of various concentrations of mannitol, sucrose, trehalose, and gelatin in 10 mM phosphate buffer. Ultimately, we found that lyophilization in 10 mM phosphate buffer, containing 1% *w/v* sucrose, 1% *w/v* mannitol, and 0.25% *w/v* gelatin, resulted in an approximately 50% decrease in viability immediately following lyophilization and reconstitution which was considered to be an acceptable outcome (Figure 6). Recovery was similar when *ΔclpB* prepared using this formulation was stored at −20 or −80 °C for more than 3 years (Figure 6). In contrast, storage at +4 °C, for the same period of time, resulted in an additional 50% decrease in viability. In contrast, no viable bacteria were recovered after storage at an ambient temperature for <6 months (not shown). Viability was similar whether *ΔclpB* was lyophilized neat (10^10^ CFU/vial) or a 1:1000 dilution (10^7^ CFU/vial), as shown in Figure 6. 

### 2.5. Quality Control and Lot Release Assays for Clinical Lots of ΔclpB

Establishment of a panel of robust in-process quality control (QC) assays and lot release assays, including those used for stability testing, are an essential component of product development. We expected to include traditional QC testing applied to control consistent production of live attenuated vaccines (e.g., monitoring bacterial growth, sterility, pH, osmolarity, moisture retained in final lyophilized vials, endotoxin, and tests for excipients and residual materials of concern, as detailed in Table 6 and Appendix A. Additionally, in the current studies, samples of *ΔclpB* collected at various stages of the development process (Appendix A) were subjected to a battery of biological tests to determine whether these showed any deviations that might correlate with a loss of potency. In total, eight different experiments performed over a period of 10 years yielded comparable results with all parameters tested. Among other assessments, serum cytokine and chemokine levels four days after ID vaccination with the various *ΔclpB* samples were comparable, including when measured using different Luminex technologies (fluorescent versus magnetic beads) and instruments (Luminex Magpix vs LS100). This outcome suggests the robustness of this assay.

## 3. Discussion

The development of vaccines against tularemia was originally prompted by the need to control endemic disease. In the early half of the 20th century, *Fth* was responsible for large epidemics involving millions of cases in the former USSR [10]. In North America, tularemia caused by both *Fth* and *Ftt* was recognized as an occupational and recreational hazard [15,56,57,58]. Historically, inhalation of *Ftt* had a mortality rate of >30% before the antibiotic era [1], and human infection studies have demonstrated that inhalation of as few as 20 CFU of *Ftt* caused severe infection that required intervention with streptomycin [12]. The incidence of tularemia worldwide declined dramatically during the latter part of the 20th century, likely related to changes in housing and work patterns, and antibiotic development made tularemia a treatable disease when promptly diagnosed. More recently, tularemia vaccines have been of interest due to the biodefense considerations. Nonetheless, some areas of the former Soviet Union continue to use live attenuated vaccines to respond to tularemia outbreaks [59], and vaccination may be of interest here and in areas such as central Sweden and Turkey [60,61], which suffer episodic disease. Therefore, vaccines such as *ΔclpB* may be useful as a biowarfare deterrent and response, and for public health purposes.

Standards and expectations for biological product development, including new vaccines, have changed dramatically since LVS was derived and human studies initiated in the 1960s. Currently, most new vaccines are subunit formulations, comprised of pathogen components and adjuvants. These approaches are most applicable to viral and bacterial pathogens for which induction of specific antibodies and memory B and T follicular helper (Tfh) cells dominate protective mechanisms, as reviewed in [62]. In contrast, relatively few vaccines have been developed against intracellular bacteria such as *Francisella*. For a variety of reasons, some of which remain poorly understood, live attenuated vaccines have held the most promise for providing substantial and durable protection against this class of microbes [63]. For example, *Fth*-derived Strain 15 was used in mass vaccination campaigns in the USSR that ameliorated large natural tularemia outbreaks [10,64]. However, because growth conditions may alter bacterial properties and because terminal sterilization is not possible for a live bacterial product, production of live attenuated vaccines presents unique challenges. Moreover, by definition, live attenuated strains pose inherently more risk for human use than static vaccines, inviting scrutiny that necessitates additional product characterization.

The difficulties in advancing vaccine candidates to market is perhaps illustrated by the experience with LVS, which remains unlicensed despite decades of study. Strain 15, one of the predecessors of LVS, and LVS itself, have been developed empirically by repeated passages in vitro and in vivo [9,65]. A lack of understanding about the genetic basis for attenuation has been accompanied by concerns about reversion of LVS to wild-type *Francisella*, and underlying genetics were not understood until the development of sequencing methods that allowed complete evaluation of the entire bacterial chromosome. This concern has largely been alleviated by subsequent genetic studies [27], and DVC’s efforts supported by NIH have produced new lots of LVS for human use by better controlled manufacturing procedures [18,66]. Nonetheless, to the best of our knowledge, no party is currently interested in advancing LVS toward licensure.

Other new tularemia vaccine candidates include the use of killed or subunit protein vaccines with various adjuvants, heterologous attenuated recombinant vaccines, and attenuated mutants of LVS, *F. novicida*, *Ftt*, and *Fth*. The properties and relative merits of these have been reviewed elsewhere [67]. We propose that *ΔclpB* provides a genetically stable, well-defined alternative to LVS. The focus on *ΔclpB* was based in part on the observation that a natural mutant of SCHU S4, *Ftt* strain FSC043, which contains a similar genetic lesion as LVS, was at least as attenuated for mice as LVS but protected mice better against aerosol challenge with *Ftt* strain FSC033 [24]; notably, FSC033 is a clinical isolate that is more virulent than SCHU S4 in rodents [68]. This may be because *Ftt* expresses several immunogenic antigens that are lacking in *Fth* strains, some of which could provide additional protection against the most virulent *Ftt* strains as compared with LVS. This hypothesis has also been borne out by data involving various mutants of *Ftt* and *Fth* lacking the *ΔclpB* gene [25]. Most recently, extensive efficacy studies using rats and cynomolgus macaques have demonstrated that *ΔclpB* provided excellent and durable protection against aerosol challenge with *Ftt* SCHU S4 (manuscripts in preparation).

An emphasis has been placed on developing live vaccines, which contain two attenuating mutations, to decrease risk of reversion. However, data following initial screening indicated that *ΔclpB* was an excellent candidate in terms of both safety and efficacy (Table 1, Table 3 and Table 4, and Figure 3 [25,29,38,69]), but further deletions impaired its efficacy potential [25]. Instead of seeking double mutants, therefore, we focused on extensive safety-related characterizations of *ΔclpB*, as detailed here. In vitro passage studies, including co-culture studies to mimic the worst-case scenario of recombination with virulent *Francisella* in nature (Table 3, Figure 3), support the conclusion that the risk of revision of this large deletion is below any limit of detection. Studies in vivo using mice, guinea pigs, and rats (Table 2, Figure 2 and Figure 4) further support a very high level of *ΔclpB* attenuation. We considered clearance studies using rats to be particularly important. Although evaluation of some tissues was limited by microbiota-related overgrowth, rat spleens were uniformly free of contaminants. In mice, the spleen is the last of the organs to clear LVS, *ΔclpB*, and other mutant *F. tularensis* strains [29,37,70]. If the same holds true for rats, then, splenic clearance is the most important tissue to monitor.

A package including all safety-related data described was submitted to the CDC for review, and the request to exclude *ΔclpB* from the Select Agent list was granted in November 2014. The same data package was sent to the Public Health Agency of Canada (PHAC), and *ΔclpB* was reclassified to containment level 2 (CL2/BSL2) for in vivo and in vitro uses, including large-scale production. Additionally, based on the relative virulence for mice of *ΔclpB*, the PHAC also reclassified SCHU S4 *ΔwbtC*, *ΔwbtI*, *ΔkdtA*, *glpX*, and *ΔlpcC* mutants as CL2 pathogens (https://health.canada.ca/en/epathogen, accessed on 27 May 2021). Since the approvals from CDC and PHAC, we further evaluated the stability of the *ΔclpB* genome at various stages of the development process described herein. We found that no genomic changes occurred as a result of the various manipulations used in the manufacture of this mutant strain; demonstrated the lack of detectable contamination of *ΔclpB* research lots by exogenous bacteria; and demonstrated the lack of detectable lytic or lysogenic phages from *ΔclpB* stocks used in all stages followed for the GMP production of *ΔclpB* (data not shown). Collectively, the available data support the substantial degree of irreversible attenuation of *ΔclpB*.

Production studies described here (Table 4, Table 5 and Table 6, Figure 5) clearly demonstrate successful scale-up. Of note, most studies described here were performed with *ΔclpB* produced at a 20–22 L scale, sufficient for millions of doses. Equally important, experience with LVS and now with *ΔclpB* indicate that lyophilization conditions have been defined that allow for long-term storage (Figure 6). This is an important feature for inclusion in stockpiles held for biodefense purposes. Given the excellent preservation of viability of *ΔclpB* in the medium developed here for a minimum of 3 years and given the prolonged preservation of lyophilized LVS following an initial 90% decline in viability, we anticipate successful storage of the former at +4, −20, or −80 °C could lead to multi-decade preservation similar to that observed with LVS.

Working quality control tests to evaluate newly produced lots as well as stability have been established and continue to be refined. Our research and product development experience, to date, further suggests several additional approaches for *ΔclpB*-specific identity, safety, and potency tests. For example, identity testing by qRT-PCR based on deletion of the *ΔclpB* gene and the expected sequences of the surrounding bacterial chromosomal changes has been established and qualified by Ology Biosciences. Understanding of the residual virulence and behavior during infections of mice and rats (Table 1, Table 2, Table 3 and Table 4) suggests options for initial safety testing that may ultimately be replaced by in vitro measures, including sequence data.

Arguably, the most critical test performed is potency testing, and its design merits careful consideration. For many years, potency testing of LVS has relied on parenteral vaccination of mice, which allows for assessment of residual vaccine virulence, followed by parenteral challenge with fully virulent *F. tularensis*. Ideally, future options will move away from extensive use of animals, particularly one such as this that depends on lethal challenges with BSL-3 Select Agents. Full development of potency testing should include consideration of a stressed preparation of *ΔclpB* that exhibits reduced protection of animals against respiratory challenge with fully virulent *F. tularensis* SCHU S4. Relevant measurements, some of which avoid lethal challenge in vivo, may include tests that have been used to evaluate multiple different iterations of *ΔclpB* over time. For example, original *ΔclpB* stocks, research production lots, lyophilized lots, and stored lots have been tested for in vivo virulence in mice (IN LD_50_) and for protection against IN challenge with fully virulent *F. tularensis*, analogous to LVS potency testing (Appendix A). Clinical scores after vaccination, skin reactogenicity, bacterial genomes (Appendix A), and levels of cytokines in sera within 4 days after vaccination (Appendix A) have been monitored, all options that do not depend on lethal infections. Moreover, we have extensively explored the use of an in vitro co-culture assay that measures the ability of lymphocytes from *ΔclpB* vaccinated mice or rats to control the intramacrophage growth of *Francisella* bacteria [46,54,71,72,73,74]. The degree of bacterial growth control tracks well with the degree of in vivo protection [54,73,75], making this assay a potential potency assay. Only small numbers of vaccinated animals are required, and macrophages can be infected with BSL-2 strains of *F. tularensis* (including *ΔclpB* and LVS), eliminating the need for BSL-3 biocontainment.

Taken together, to date, studies of *ΔclpB* indicate that it is an excellent tularemia vaccine candidate. The stable, high degree of attenuation by many criteria coupled with its strong efficacy profile yield a favorable risk/benefit relationship. Moreover, commercial production at scale is feasible and economical. These factors support completing animal studies coupled with derivation of immunological correlates of vaccine-induced protection, in anticipation of proceeding to human clinical trials.

## 4. Materials and Methods

### 4.1. Generation of ΔclpB and Evaluation of Its Potential for Reversion to Wild Type

Mutant *ΔclpB* was created by deletion of 2463 out of 2580 bp of the *clpB* gene using technology developed at Umea University [26]. Complementation was performed, as previously described [76]. Tests for reversion included passage of *ΔclpB* through mice without intervening plating. An initial mouse was injected intraperitoneally (IP) with *ΔclpB* and was euthanized on Day 3 of infection; its spleen was removed, homogenized, and re-injected into a second mouse. Mice 3–5 were similarly infected, for a total of 5 passages without intervening plating. Then, a stock of in vivo passaged *ΔclpB* was prepared by plating a 1/1000 dilution of the spleen from Mouse 5 on cysteine heart agar supplemented with hemoglobin (CHAH). The resulting bacterial lawn was used to make liquid suspensions that were aliquoted in small volumes and frozen at −80 °C, as previously described [24]. Additionally, we passaged *ΔclpB* 10 times daily on CHAH agar, and then made stocks of the final passage, as above. Finally, to evaluate general stability and protein production of the bacterial mutants, we co-cultured *ΔclpB* and SCHU S4 *ΔiglC* on CHAH through 5 daily passages in vitro. Assuming a 2-hour doubling time, this represents at least 60 doublings, or a >1 × 10^18^-fold increase in viable bacteria over the initial inoculum. Bacteria from the fifth passage were lysed, and the presence of clpB and iglC proteins was demonstrated by mass spectrometry in approximate proportions to the quantities found in the individual mutants using previously described methods [69]

### 4.2. Fermenter Growth of ΔclpB at NRC-C

Fermenter growth of *ΔclpB* used a working volume of 22 L in a 30 L new MBR Vessel (Multiple Bioreactors and Sterile Plants AG, Zurich, Switzerland). All fermenter parameters were monitored and controlled by custom software. As proof-of-principle, we first tested large-scale *ΔclpB* growth in Chamberlains defined broth (CBD) [44]. A 1 mL vial of a frozen stock of *ΔclpB* that had been passaged 5 times in CBD was thawed and transferred to a 4 L baffle flask containing 1 L of CMB. After overnight incubation at 37 °C with shaking, sufficient starter culture was added to a fermenter vessel containing 22 L of sterile CBD to give a final OD_600_ reading of 0.1. At 16, 19, and 21.5 h, 150 mL samples were sterilely withdrawn from the fermenter, sucrose added to 20% *w/v*, and then 1.0 mL aliquots were dispensed and frozen at −80 °C. Thawed aliquots of frozen stocks were used for testing in mice. Subsequent runs were performed as above using the modified casein partial hydrolysate broth described by Karlsson et al. [77].

### 4.3. Manufacturing of ΔclpB at the 25 L Scale at Ology Bioservices

To make a master cell bank, single isolated colonies of *ΔclpB* were grown and selected on buffered charcoal yeast extract (BCYE) agar plates. Then, single colonies were expanded in MCPH broth. Production runs were performed under Current Good Manufacturing Procedures (cGMP) and documented with Master Batch Records. Seed strains were established by inoculating 1 L shake flasks containing 250 mL of MCPH broth. Cultures were incubated for up to 24 h at 37 °C, shaking at 200 RPM until the OD_600_ reached 1.5 ± 0.3. Samples of the broth were evaluated for CFU/mL prior to freezing. Bacteria were aliquoted as 1 mL in 2 mL cryovials. The bacteria were frozen at <−70 °C in a solution of 1 g/L sucrose in MCPH media. The *ΔclpB* production runs were performed as follows: *ΔclpB* seed strains were established by inoculating 1 L shake flasks containing 250 mL of MCPH broth with a single vial of the research cell bank. Cultures were incubated for up to 24 h at 37 °C shaking at 200 RPM, until the OD_600_ reached 1.5 ± 0.3. The fermentation runs were performed by inoculating 24.5 L of MCPH broth with a dilution of the shake flask culture, to reach an initial OD_600_ = 0.0015 with a total of 25 L of MCPH broth in a 50 L XDR-50 fermenter (Cytiva, Marlborough, MA). The fermentation process was conducted by incubating the cells at 37 °C for approximately 20 h. 

### 4.4. Lyophilization of ΔclpB Drug Product

At NRC-C, small volumes of *ΔclpB* (50–100 mL) were buffer exchanged with lyophilization buffer (10 mM potassium phosphate, 1% mannitol, 1% sucrose, 0.25% gelatin, pH 7.2.) by centrifugation, washed, and resuspended to the original volume. The GMP *ΔclpB* drug product from Ology Biosciences was lyophilized by Lyophilization Technology Inc. (Warminster, PA, USA). In this case, *ΔclpB* was buffer exchanged by diafiltration. Thereafter, rinsed 2 cc/13 mm vials were filled to a target volume of 0.50 mL, and then lyophilized using a proprietary cycle program. Then, final vialed drug product samples were characterized for release testing.

### 4.5. Analytical Assays

Colony forming units were determined by plating serial dilutions of *ΔclpB* on CHAH, BCYE, or supplemented chocolate agar plates (as available, with equivalent results) that were incubated at 37 °C for 48–72 h. Antibiotic resistance was performed by plating *ΔclpB* onto peptone cysteine agar plates and incubated for 72 h at 37 °C. Antibiotic discs containing each antibiotic were placed on the surface of the bacterial lawn and the plates incubated for 48 h at 37 °C. The resulting zone of complete inhibition was measured in millimeters. Endotoxin testing was performed using the Limulus Amebocyte Lysate Endosafe nexgen-PTS system, following the manufacturer’s instructions (Charles River Laboratories, Wilmington, MA, USA). Whole genome sequencing was performed by Omega Bioservices (Norcross, GA, USA) and in-house by NRC-C. A PCR-based method for identification of the deletion mutant was developed using the Phusion Flash Master Mix (Thermo Scientific, Waltham, MA, USA). Two sets of primer were designed. The first set of primers were based on the sequences outside of the *clpB* gene and the other set included sequences from within the gene. Thirty-five cycles of the PCR reaction were completed followed by a five-minute elongation step.

### 4.6. Biological Assays

Samples of *ΔclpB* generated at various stages of the product development cycle were subjected to a battery of in vivo tests in young adult female BALB/c mice. These assays included in vivo organ growth, ID and IN virulence determinations, clinical scores and skin reactogenicity, potency against ID or IN challenge with SCHU S4 or FSC033, and serum cytokine production after ID vaccination with 10^5^ CFU (Appendix A). 

## 5. Patents

Some of the work published herein is the subject of patents US8993302B2, EP2424974B1, CA2760098C, ES2553763T3, US20210008191A1 (pending), CA3094404A1 (pending), EP3768820A1 (pending).

## Figures and Tables

**Figure 1 pathogens-10-00795-f001:**
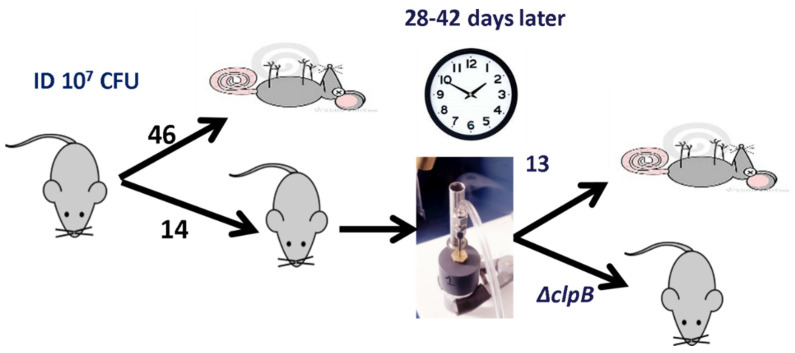
Mouse screening protocol for identifying SCHU S4 mutants with vaccine potential against respiratory challenge with fully virulent *Ftt*. For initial screening, mutants were tested for virulence at an ID inoculum of 10^3^ CFU in young adult female BALB/c mice. Then, mutants that caused no overt signs of illness were inoculated ID at 10^5^ or 10^7^ CFU. Mice that survived the 10^7^ ID dose were then exposed to a low dose aerosol (~100 CFU) of either *Ftt* strain FSC033 or SCHU S4 and were monitored for survival. *ΔclpB* was the only mutant that was able to provide protection (survival and prolonged time to death) against aerosol challenge.

**Figure 2 pathogens-10-00795-f002:**
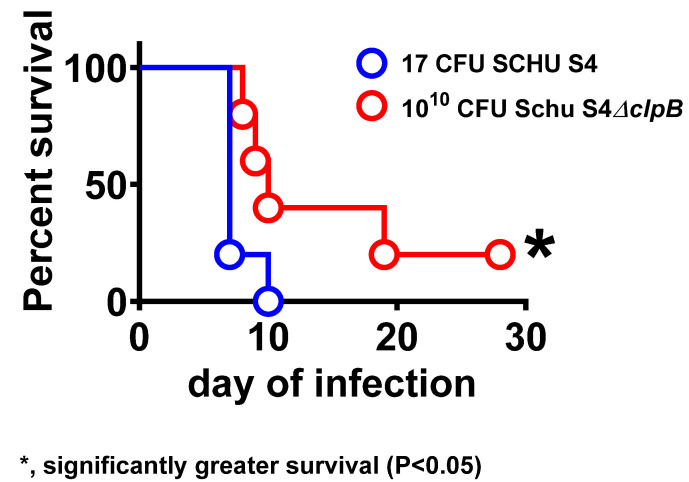
Attenuation of *ΔclpB* for guinea pigs. Guinea pigs, which are as susceptible to virulent *Ftt* challenge as mice, were administered the indicated doses of SCHU S4 or *ΔclpB* SC and monitored for morbidity and mortality to further evaluate the attenuation of *ΔclpB*. Whilst a dose of 17 CFU of SCHU S4 administered SC killed all guinea pigs within 10 days, 10^10^ CFU *ΔclpB* led to significantly lower morbidity and mortality.

**Figure 3 pathogens-10-00795-f003:**
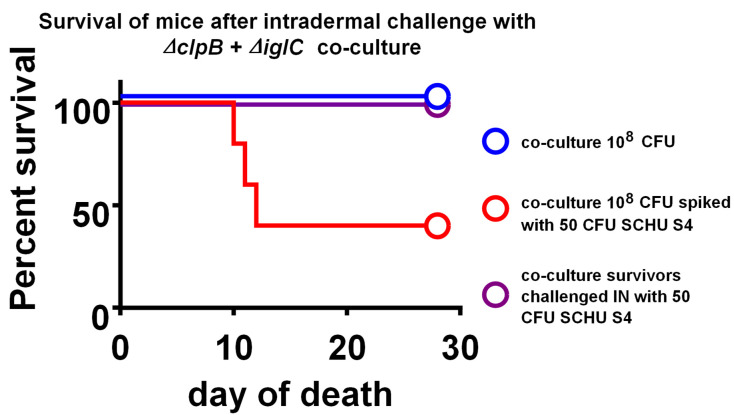
Failure of *ΔclpB* to revert to a virulent phenotype following prolonged co-culture with another highly attenuated SCHU S4 *ΔiglC* mutant. The mutants were both co-cultured on CHAH agar for 48 h through five passages, as described in Materials and Methods. Thereafter, proteomics (2D PAGE followed by mass spectrometry) confirmed the presence of the clpB and iglC proteins at levels expected from either mutant alone. BALB/c mice were administered the indicated CFU of co-cultured bacteria ID and monitored for survival. Co-cultured bacteria remained as attenuated for BALB/c mice as the individual mutants, whereas spiking the co-culture preparation with as few at 50 CFU of wild-type SCHU S4 was sufficient to cause death. Co-cultured bacteria also engendered protection; following administration of co-cultured bacteria, surviving mice were also challenged with 50 CFU SCHU S4, and all survived.

**Figure 4 pathogens-10-00795-f004:**
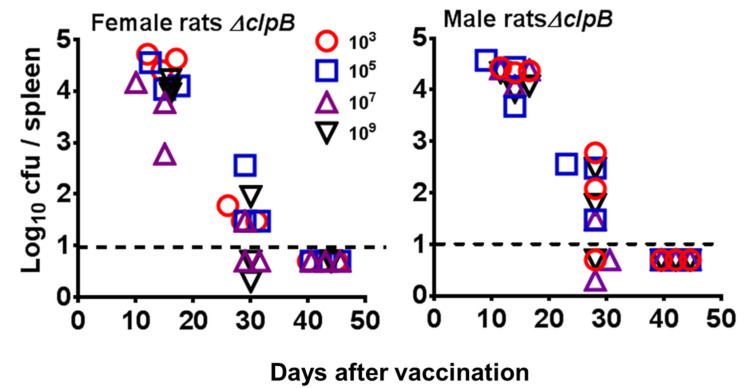
Clearance of *ΔclpB* from the spleens of vaccinated Fisher 344 rats. Young adult male and female Fischer rats were inoculated ID with the indicated doses of *ΔclpB*. At the indicated times after inoculation, rats (*n* = 3/group) were euthanized and the splenic burdens of *ΔclpB* determined. As illustrated, rats of either sex cleared *ΔclpB* between 30 and 42 days after vaccination. Dashed horizontal line shows limit of detection.

**Figure 5 pathogens-10-00795-f005:**
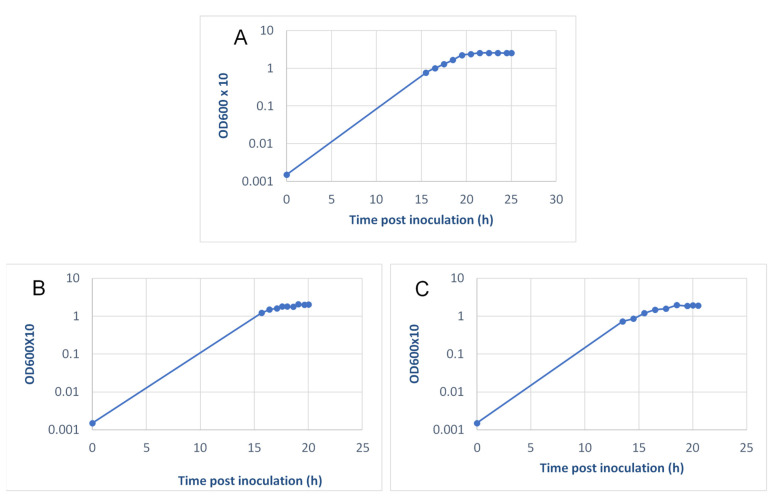
Reproducibility of the 25 L manufacturing process. *ΔclpB* was cultured as described in Materials and Methods. Aliquots were sampled from approximately 15 h onwards and growth was monitored by OD_600_; samples were diluted 1:10 prior to reading. The line from 0 to 15 h is stylized from the fermenter printouts to avoid potential contamination of the fermenter vessels by sampling at low *ΔclpB* levels. (**A**) NRC-C Run 2); (**B**) ology process Run 1; (**C**) ology process Run 2.

**Figure 6 pathogens-10-00795-f006:**
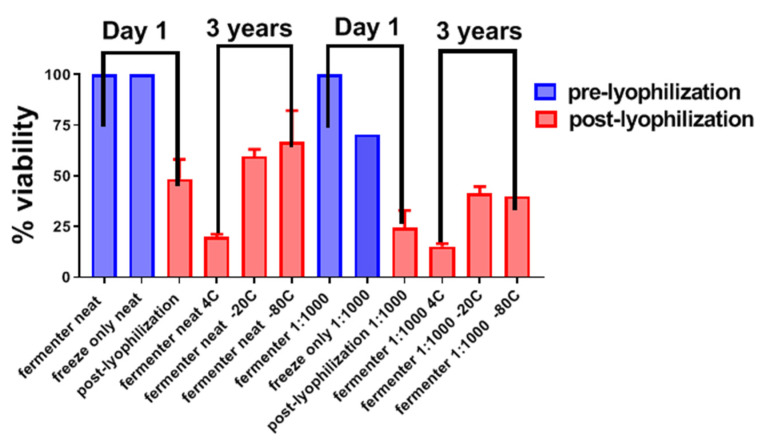
Viability of *ΔclpB* following lyophilization and prolonged storage at different temperatures. *ΔclpB* was obtained from a fermenter run, and the suspension buffer was exchanged with lyophilization matrix by centrifugation and washing twice with matrix. At high concentration (~10^10^ CFU/mL), freezing the samples prior to lyophilization had no impact of the viability of *ΔclpB*, but at a lower concentration (~10^7^ CFU/mL), this was reduced by ~25% (not shown). On Day 1 following lyophilization, the viability of *ΔclpB* decreased ~50% and ~75% at the higher and lower concentration, respectively. Next, lyophilized *ΔclpB* was stored for 3 years at +4 or −20 °C or −80 °C. Lyophilized *ΔclpB* stored at +4 °C only decreased further in viability to ~50% at this time point.

**Table 1 pathogens-10-00795-t001:** Virulence properties of SCHU S4 deletion mutants for BALB/c mice.

ID Virulence for BALB/c Mice of 60 Distinct Single Gene Deletion Mutants of SCHU S4		
ID LD_50_ <30 CFU^a^ *ΔFtt*	ID LD_50_ >10^2^ but <10^6^ CFU *ΔFtt*:	ID LD_50_ ≥ 10^7^ CFU *ΔFtt*
*ahp* ^b^ *, capB, chiX, fadAD2, fadD2, feoB, gpx, hfq, katG, mdaB, oppD, oxyR* ^c^ *, pckA-L* ^d^ *, pdpC* ^c,e^ *, pdpD, pepO, PI1* ^c,g^ *, PI2* ^b,g^ *, PilAEV ^f^, pilB, pilC, pilQ, PilT, pmrA* ^c^ *, pyrB, qseC* ^b^ *, RD5, RD8, recA, relA, rimK, sodC, tet, usp, 0023, 0024, 0029, 0069, 0086, fupB, 1023* ^c^ *, 1149* ^b^ *, 1564* ^b^	*pckA-S* ^d^ *, fupA* ^c^ *, ggt* ^f^	*iglB, iglC, iglD, gplX, wbtC, wbtI, clpB* ^f^ *kdtA, lpcC, purF, sspA, mglA, FLT0439* ^g,h^ *FTT0085,*

^a^ Actual inocula ranged between 5 and 30 CFU; ^b^ extended time to death compared to SCHU S4; ^c^ ≤40% survival; ^d^
*pckA* produced large (-L) and small (-S) colony types; ^e^ single gene copy deleted; ^f^ 60–100% survival against respiratory challenge (~100 CFU IN or aerosol with SCHU S4 and/or FSC033); ^g^ deletion spanned several adjacent genes; ^h^ gene designation describing deletion found in LVS. For FTT designations see Appendix A).

**Table 2 pathogens-10-00795-t002:** Comparisons of LVS and SCHU S4 *ΔclpB* undertaken over the past 22 years (LVS) or 14 years (SCHU S4 *ΔclpB*) at the National Research Council of Canada (NRC-C) that formed part of the Select Agent removal request.

Criterion	LVS	*ΔclpB*	FSC033 ^a^, SCHU S4 ^a^ (MTD) ^b^
ID LD_50_	>10^7^ CFU	>10^7^ CFU	<10 CFU (7)
IN LD_50_	~10^3^ CFU ^c^	10^4^–10^6^ CFU ^d^	<10 CFU (6)
% deaths following ~10^5^ CFU ID ^e^	2.1 (*n* = 185) ^f^	1.4 (*n* = 560) ^f^	100
Necrosis score at site of injection	2–3 ^g^	0–1 ^g^	4 ^g^
SCID mice MTD ID 10^3^ CFU ID	15	17 ^h^	ND
SCID mice MTD IN 10^2^ CFU IN	14	19	ND
Clinical signs given at 10^5^ ID	Mild	Mild	severe
Survival SCHU S4 20–100 CFU IN ^i^	≤20%	60–100% ^i^	NA

^a^ Two distinct virulent strains of *Ftt*; ^b^ (MTD) = median time to death; ^c^ ~1 × 10^3^ CFU of LVS IN invariably killed all BALB/c mice over a 20 year span at NRC-C; ^d^ inter- experimental range; ^e^ normal ID vaccination dose; ^f^ not always head-to-head comparisons; ^g^ score range 1–6 (see Golovliov et al. [25]; ^h^ significantly longer survival as compared with LVS; ^i^ survival 42 days after immunization followed by IN challenge with 20–100 CFU of SCHU S4.

**Table 3 pathogens-10-00795-t003:** Effect of culture conditions on attenuation of original *ΔclpB*.

Source of ΔclpB	% Survival after IN Inoculation with ~104 CFU	% Survival after IN Challenge with ~100 CFU SCHU S4 42 Days Later
Original stock	100%	100%
10× in vitro ^1^	100%	100%
5× in vivo ^2^	100%	100%
Original stock	100%	100%
5× in CDM ^3,4^	60%	100%
Fermenter CDM ^5^	40%	100%

^1^ Original stock passaged by serial streaking for confluent growth on CHAH followed by 48 h incubation at 37 °C, final passage was resuspended in freezing medium and used herein [24]; ^2^ serial passage of infected spleens by IP inoculation without intervening in vitro passage, spleen from the 5th passage was grown to confluence on CHAH agar, resuspended in freezing medium, and used herein; ^3^ CDM, Chamberlains defined broth [44]; ^4^ passaged 5× in CDM for acclimation prior to ^5^ fermenter growth.

**Table 4 pathogens-10-00795-t004:** Characterization of *Δ**clpB* used to produce a research cell bank.

Characteristic	
Time of harvest from flask	20.75 h
CFU/ml at harvest ^a^	3.9 × 10^9^
Cryopreservative	Sucrose 10% *w/v*
Colony morphology ^b^	Typical of *Ftt* and *Fth*
Gram stain ^c^	Typical of *Ftt* and *Fth*
Lytic phage	Negative
Lysogenic phage	Negative
Contaminant bacteria ^d^	Negative
Genomic sequence ^e^	Identical to original mutant
42-day survival of mice after 10^4^ CFU IN administration (original stock/new stock) ^f^	100%/60% at day 42; NS ^h^
42-day survival of mice following 10^5^ CFU ID administration (original stock/new stock)	100%/100%
Clinical signs ^g^ (original vs. new stock)	Significantly greater vs. old stock on days 3, 4, and 5
Skin reactogenicity ^g^ (original vs new stock)	NS ^h^
*ΔclpB* organ load (skin, spleen, liver, lung) 4 days after ID vaccination with 10^5^ CFU of original vs. new stock.	*ΔclpB* lung burden was significantly higher (*p* = 0.016) for new vs. original stock
Day 4 serum cytokine/chemokine levels (21-plex assay) (original vs. new stock)	NS ^h^
28-day survival against IN challenge with ~100 CFU SCHU S4 after IN vaccination (original/new stock)	100%/100%
Protection against IN challenge with ~100 CFU SCHU S4 after ID vaccination original/new stock (MTD in days)	60% (28)/0% (16). NS ^h,i^
Protection against ID challenge with 10^5^ CFU SCHU S4 or FSC033 after ID vaccination	100%/100%

^a^ Average from plating of 2 vials each by three individuals; ^b^ after 48 h growth on Oxoid chocolate II agar; ^c^ no evidence of contaminating bacteria; ^d^ several vials were plated on regular Oxoid chocolate agar or sheep blood agar, and incubated aerobically, microaerophilically, or anaerobically at 37 °C for 72 h; ^e^ no evidence of any contaminating DNA; ^f^ inocula for old vs. new stock were 7 × 10^3^ CFU vs. 1.32 × 10^4^ CFU; ^g^ scored blind by NRC-C animal resources staff; ^h,^ not significantly different from each other; ^i^ when compared to naïve mice (MTD 5 days) and corrected for multiple comparisons.

**Table 5 pathogens-10-00795-t005:** Comparison of *ΔclpB* yields from 22 L fermenter runs at NRC-C in MCPH broth.

Date of Run	Flask Starting Inoculum CFU/mL	Flask Harvest Concentration CFU/mL (Culture Time, h)	#Doublings (Doubling Time, h)	Fermenter Inoculum CFU/mL	Fermenter Harvest Time, h (CFU/mL)	#Doublings (Doubling Time, h)	Total CFU × 10^14^ (# Doses × 10^6^) ^1^	Biomass g
December 2014	4.0 × 10^7^	5.2 × 10^9^ (18.5)	7 (2.6)	3.90 × 10^6^	22 (1.1 × 10^10^)	12 (1.85)	242 (24)	177.3
September 2015	2.6 × 10^7^	9.0 × 10^9^ (18.25)	8.4 (2.1)	1.40 × 10^7^	20.5 (1.1 × 10^10^)	10 (2.3)	242 (24)	204.1
June 2017	7.8 × 10^6^	9.0 × 10^9^ (18.5)	10.2 (1.8)	1.43 × 10^7^	26 (1.1 × 10^10^)	10 (2.9)	242 (24)	74.9 ^2^

^1^ # of human doses based on an inoculum size of 10^7^ CFU/ml as used for LVS; ^2^ remaining biomass after periodic sampling.

**Table 6 pathogens-10-00795-t006:** Quality control testing of the final drug product.

Parameter	Test Method	Specification
	*clpB* specific PCR	Identity confirmed ^a^
	Purity	Absence of contaminating organisms ^b^
	CFU/mL	Report result
	Gram Stain	Gram negative coccobacillus
	TBD ^c^	Round, smooth and slightly mucoid single colonies
	CFU/Ml ^d^	Report result
	LAL ^e^	<5 EU/kg/dose
	Susceptible to tetracycline, levofloxacin, gentamycin, chloramphenicol, ciprofloxacin, streptomycin, rifampin ^f^	Susceptible
	Karl Fisher ^g^	Report Result
	Appearance of cake ^h^	Report result

^a^ Whole genome sequencing was performed on Master Cell Banks, the PCR method developed to demonstrate the absence of intact *clpB* gene is described in methods section; ^b^ growth on multiple media known to support and inhibit the growth of *ΔclpB* ; ^c^ bacteria were grown on chocolate agar plates; ^d^ Bacteria were counted prior to placing into a known volume of media, samples were taken at various times, plated on agar plates, and colonies counted; ^e^ LAL = limulus amoebocyte lysate assay using an Endosafe nexgen-PTS system (depending on the safety margin required by the regulator, the required test dose of *Δ**clpB* could fail this test because it possesses intact LPS albeit with low endotoxicity [51], ^f^ by antibiotic disc assay on chocolate agar medium; ^g^ for dryness of lyophilized cake; ^h^ color and form as compared with a standard sample, ease of solubilization, residual solids after reconstitution.

## Data Availability

The data presented in this study are all available in the present article.

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
