# Peer review of "Modern Development and Production of a New Live Attenuated Bacterial Vaccine, SCHU S4 ΔclpB, to Prevent Tularemia"

_pathogens, 2021, doi:10.3390/pathogens10070795_

Round 1
Reviewer 1 Report
This manuscript is part review, and part publication of original research which details the steps and data supporting the manufacture and evaluation of a promising live attenuated vaccine for tularemia. It has broad relevance to the vaccine and bio-threat research fields and helps educate the research community on the steps and challenges of translational vaccine research.
General Comments;
- Results, line 145. Was a dose of ΔclpB determined that showed no lethality in guinea pigs and how did this compare to relative virulence of LVS in guinea pigs?
- Results, line 321. Long term storage of lyophilised material at 4oC resulted in an additional 50% loss of viability compared with storage at <-20o Yet, data in Table S4 demonstrates no loss of efficacy. In fact, vaccine preparations stored at <-20oC had reduced efficacy in comparison. Can you comment on what appears to be a counter-intuitive result.
- Results, line 345. The authors highlight the use of a ‘battery of biological tests to determine whether these showed any deviations that might correlate with loss of potency’ (Tables S3, S4). The measurement of cytokine responses is also highlighted in the Discussion. It is inferred that these may be useful for potency testing. Whilst I support the measurement of these parameters, there appears to be little correlation between the magnitude of cytokine responses for different production batches in Table S3 to efficacy reported in Table S4. I would therefore caution against the over extrapolation of this data with regards to potency.
- Has any view been provided, or sought, from representatives of regulatory bodies on their acceptability of a live attenuated strain with only a single gene deletion? Is there an expectation that the safety data provided will meet regulatory requirements and/or do regulators set defined safety bench marks for live attenuated vaccines?
Minor Type Corrections;
- Consistency in using SCHU S4 rather than Schu S4 as used in title and in Figure 2 legend.
- Discussion, line 472. Comment refers to Table S3 not S4.
- Discussion, line 475. Comment refers to Table S4 not S3.
Author Response
Pleae see attachment

Reviewer 2 Report
Major comments:
Table 1. First, it is unusual to have a summary Table in the introduction, especially given that reference 28 does not include all of the mutants described in Table 1. All appropriate references should be included in Table 1 and authors should consider moving Table 1 to Results. Second, readers would greatly benefit from having gene locus numbers being listed next to gene names. Third, some gene names in the left column are incorrectly capitalized. Fourth, some gene loci do not have FTT, FTL, or FSC in front of their locus numbers (column 1) so readers will not know which subsp and strain locus is being referenced. Fifth, clpB (line 97) does not appear to be listed in Table 1. Sixth, glpX is described on p.3, line 97, but gplX is listed in Table 1. There are many other inconsistencies and errors in Table 1 that should be corrected (e.g. Ftt043, LpCC, FLT0439).
p.5, Table 2. It is confusing to have superscripts for both scientific numbers and footnotes (e.g. IN LD50, ~10e3 CFUe3…CFUe4. Also Table 3, right column has two numbers, (7) and (6) in parenthesis. What are these? Suggest changing table footnotes to superscript letters or symbols (#, $, %, etc.)
p.5, Table 2. It appears that the point of this paper is to highlight that the clpB mutant is attenuated and therefore safe to be used as a vaccine. For the ‘Growth in human THP-1 macrophages,’ without any comparable data for either FSC033 or SCHU S4, it is difficult to assess if the clpB mutant is attenuated or not. Please include FSC033/SCHU S4 data or remove LVS and clpB mutant data.
p.6, line 155-156. Authors note “10e4 CFU (two logs lower than ΔclpB’s starting LD50; Table 2).” However, Table 2 notes that the LD50 of clpB is 10e4-10e6. This is not two logs lower than the starting LD50 but, rather, the lowest LD50 dose.
p.6 line 173 and Figure 3 legend. Line 173 notes ‘bacteria spiked with 50 CFU SCHU S4’ and ‘inoculation of 50 CFU of SCHU S4 alone” but Figure 3 legend notes “as few at 10 CFU of wildtype SCHU S4…’ Please make consistent
p.7, lines 201-207 appear to be descriptions of failed experiments (contaminated skin, contaminated lungs, contaminated livers) that do not contribute to our understanding of clpB attenuation in rats. Suggest deleting.
Table 4. ’42 day survival of mice after 10e4 CFU IN challenge.’ Does this mean survival of mice 42 days after being immunized with clpB and IN challenged with SCHU S4? Similar question for ’42 day survival of mice after 10e5 CFU ID challenge? Should ‘challenge’ be changed to ‘immunization’ for both? These are not clear.
line 266 and Figure 5. Authors note ‘logarithmic and stationary phases’ but it is difficult to tell from Figure 5A if growth actually is logarithmic given clustered data from ~15-25 hours but lack of data from 0h to 15h. Not sure this statement is necessary or justified on line 266.
lines 324-325 and Figure 6. Lines 324-325 note 10e10 or 10e7 concentrations but Figure 6 lists ‘neat’ and ‘1:1000.’ Please make consistent. Authors also note that viability was similar at either concentration but Figure 5 appears to show viability of ~65% for 10e10 at -80 vs. viability of~40% for 10e7 at -80. These are not similar. Finally, please correct horizontal blue lines and bootstrapping in Figure 6, which obscures data.
Minor comments:
p.1, lines 6-13. Superscript numbers in front of affiliations are missing.
p.3, line 95 and Table 1. It appears that 61 mutants are actually listed in Table 1 (not 60 mutants as listed here and in line 26 of the abstract). As noted above, clpB appears to be missing from Table 2 so should the actual number of mutants be 62?
p.5 Table 2 legend. Acronym ‘NRCC’ should be defined
p.5 Table 2 footnote. “1, two distinct virulent…susbspecies” Subspecies is misspelled.
p.5, Table 2, Please define ‘NA’ for ‘Survival SCHU S4 20-100 CFU IN.’ Should this be ‘ND’? Suggest changing to ’Survival following SCHU S4 20-100 CFU IN challenge’
Table 2. Footnote. Superscript 8, ‘significantly longer survival compared to LVS’ does not appear to match with ‘Growth in human THP-1 macrophages.’ For this data, what is ‘Log10e8’?
Figure 2. Please remove figure title above survival curve. Figure 2 legend contains the title.
Figure 3. Please remove addition spacing between ’10’ and superscript 8. Figure 3 x axis (day of death) has a different label than Figure 2 (day of infection). Please keep consistent
Figure 4. Is the dashed line the limit of detection?
p.8, line 222. NRC-C is listed here but Table 2 noted NRCC. Please keep consistent.
Reviewer 3 Report
The manuscript by Conlan et al., describes the process used to manufacture a live attenuated vaccine for tularemia, specifically one that is more effective against pulmonary tularemia compared to the current investigational vaccine, LVS. The authors also provide the data that was used to permit removal by the CDC from the U.S. Select Agent list Francisella tularensis tularensis (Ftt) strains harboring the ClpB deletion—something that was later replicated by the Public Health Agency of Canada. The co-culture and serial passage in mice experiments are especially convincing evidence for lack of conversion to a fully virulent phenotype. The antigens responsible for conferring protection against Ftt are not well understood and many previous studies using deletion mutants of Ftt SCHU S4 have shown various degrees of success in terms of attenuation and efficacy, including studies by many of these same authors. The ability to successfully lyophilize an attenuated live vaccine such as the one described here represents a significant achievement and may permit excursions from the cold chain (such as during transport) and extended shelf-life. This reviewer finds the current manuscript to be well written and makes an important contribution to the field of tularemia, particularly when it comes to next generation tularemia vaccines that are effective against aerosolized exposure/pneumonic tularemia. Consequently, I recommend it for publication after a few minor points are addressed:
Line 153: minor typographical error in Figure 2 legend: “…109 CFU ΔclpB led to sig-nificantly lower morbidity ad mortality.” The letter ‘n’ is missing from the word “and”.
Figure 4. Clearance of ΔclpB from the spleens of vaccinated Fisher 344 rats. I find it curious that there was not an obvious dose-dependent relationship in clearance of ΔclpB from the spleens. In other words, lower doses are not cleared sooner and in fact for various timepoints up to Day 30, some of the higher cfu counts are seen for the lower doses (e.g., 103 and 105 compared to 107 and 109). Is there an obvious explanation? Is this typical? Does this reflect the inherent variability of the assay?
Table 6. Table 6. Quality control testing of the final Drug Product. A little more detail, perhaps in a footnote or in the text, would be helpful. Specifically:
- Identity: is the target specification 100% identity over the entire genome, the region containing and bracketing the deletion, or something else (e.g., >95% identity, etc.)?
- Under the column "Test Method", a brief description or the name of the actual method used should be captured. For example, Karl Fisher for residual moisture content, LAL for endotoxin. There are a few instances where the Test Method has not been captured. These include Modified microbial limits/Purity, Growth Stability, Viability, and Antibiotic Resistance.
- Modified microbial limits/Purity: absence of contaminating organisms. What methods (media, etc.,) are used to assess this and is the growth of fungal organisms, which may take longer to grow, considered?
- Morphology: on which growth medium is this assessed?
- Can the authors elaborate more on what is meant by growth stability and how this will be assessed (media used, etc.)? Only units are provided (CFU/mL).
- Antibiotic resistance--should the specification be Sensitive? Positive is potentially ambiguous. The column "Test Method" should describe the method used. In this case it is antibiotic discs containing each antibiotic placed on the surface of a bacterial lawn. The specification (pass/fail) would be sensitivity to each (and every) antibiotic.
- Is there a target specification for moisture content or a plan to develop one? Same for viability.
- Shouldn't visual appearance be included as a lot release specification? For a lyophilized DP, this could be the appearance of the lyophilized cake (e.g., elegant cake, expected color, no signs of collapse, etc.). For the reconstituted lyophilized cake, a visual appearance specification would also be useful for the reconstituted drug product and might include time to fully reconstitute.
- Endotoxin, LAL. In the footnote, it is stated that it is expected that this test will fail to meet the specification of < 5 EU/kg/dose. How can this be reconciled with the study by Sandström et al., (https://doi.org/10.1016/0378-1097(92)90064-U) wherein they state that the LPS of Fth (LVS) is relatively inert and no LPS endotoxin properties were found in the Limulus amoebocyte lysate assay? Is the LPS of tularensis tularensis (Ftt), the SCHU S4 strain or ΔclpB mutant specifically known to not be as inert as the LPS of LVS? Is there an alternative assay that can be used instead of the LAL to detect LPS from sources other than SCHU S4 ΔclpB (i.e., the drug product)? LPS could come from contaminated tubing, glassware, media, etc.
Author Response
Please see attchment
